# Gaze Estimation Network Based on Multi-Head Attention, Fusion, and Interaction

**DOI:** 10.3390/s25061893

**Published:** 2025-03-18

**Authors:** Changli Li, Fangfang Li, Kao Zhang, Nenglun Chen, Zhigeng Pan

**Affiliations:** School of Artificial Intelligence, Nanjing University of Information Science and Technology, Nanjing 210044, China; 202212620006@nuist.edu.cn (F.L.); kaozhang@nuist.edu.cn (K.Z.); chennenglun@nuist.edu.cn (N.C.); 003443@nuist.edu.cn (Z.P.)

**Keywords:** gaze estimation, convolutional neural network, feature interaction, multi-head attention

## Abstract

Gaze is an externally observable indicator of human visual attention, and thus, recording the gaze position can help to solve many problems. Existing gaze estimation models typically utilize separate neural network branches to process data streams from both eyes and the face, failing to fully exploit their feature correlations. This study presents a gaze estimation network that integrates multi-head attention mechanisms, fusion, and interaction strategies to fuse facial features with eye features, as well as features from both eyes, separately. Specifically, multi-head attention and channel attention are used to fuse features from both eyes, and a face and eye interaction module is designed to highlight the most important facial features guided by the eye features; in addition, the channel attention in the Convolutional Block Attention Module (CBAM) is replaced with minimum pooling instead of maximum pooling, and a shortcut connection is added to enhance the network’s attention to eye region details. Comparative experiments on three public datasets—Gaze360, MPIIFaceGaze, and EYEDIAP—validate the superiority of the proposed method.

## 1. Introduction

Humans use various senses to acquire information from their surroundings. The eyes, which are the primary means for external information to enter the brain, receive about 80% to 90% of the information processed by the brain. Eye movements are mainly divided into gaze, pursuit (smooth movement), and reflex, with gaze being an externally observable indicator of human visual attention. When the human eye gazes at an object for more than 0.1 s, the sensory cells on the retina have enough time to process and form a clear image [1]. Gaze estimation is a technique used to determine the observer’s gaze direction or gaze point coordinates.

Through tracking the position of the gaze of the human eye, researchers can effectively predict areas of visual attention. Accurately recording gaze positions can solve many problems and has been applied in many fields, such as user experience research [2], medical and clinical applications [3], virtual reality (VR) [4], augmented reality (AR) [5,6], assisted driving [7,8], and human-computer interaction [9].

In the current field of gaze research, some models adopt a single-stream data architecture [10,11,12,13], relying on full-face images or eye images as input, whereas others use a multi-stream data architecture [14,15,16,17,18,19], taking both as input. Clearly, the latter can produce more accurate results as it utilizes more information streams. Multi-stream models typically process data from different streams through isolated network branches to extract features and merge all features in the final stage. However, such models ignore the interaction and fusion of feature information between different data streams. Feature fusion is the process of integrating feature information from different sources, modalities, or hierarchies into a unified, more expressive feature representation. The goal is to improve model performance through combining complementary information. Therefore, these models cannot comprehensively learn and represent complex eye movement features.

To overcome the above shortcomings, this study proposes a multi-stream gaze estimation method that integrates multi-head attention mechanisms, fusion, and interaction strategies. The main contributions are as follows:An eye feature fusion module (EFFM) is designed, which uses cross-attention mechanisms multiple times to hierarchically fuse features from both eyes using fusion blocks, thereby greatly enriching the extracted eye features.A face and eye interaction module (FEIM) is designed. The facial features are weighted by the enhanced features in the binocular neighborhood to focus on eye position information, which helps in gaze estimation.In the channel attention module of CBAM, minimum pooling is utilized instead of maximum pooling, and a shortcut connection is introduced to help capture facial details.

The rest of the paper is organized as follows. Section 2 introduces some related works. Then, our motivation and method are explained. In Section 4, the experimental results of representative works are presented and analyzed, including the proposed method tested on the Gaze360 [13], MPIIFaceGaze [15], and EYEDIAP [20] datasets. Finally, Section 5 concludes the paper.

## 2. Related Work

In this section, we review related work on gaze estimation methods. Technically, gaze estimation research has gone through two stages: 3D geometric modeling and deep learning [21].

### 2.1. Three-Dimensional Geometric Modeling Stage

In the early stages, gaze estimation was achieved through 3D human eye models. In 2011, Tsukada et al. [22] introduced a 3D eye model to convert between eye position and camera coordinates. In 2015, Wood et al. [23] used head scan geometry to construct a set of dynamic eye region models. Wan et al. [24] proposed a Compact Pupil Region Detection (CPRD) method. Liu et al. [25] established the positional relationship between the optical and visual axes, obtaining gaze positions through optimization methods.

While methods based on 3D geometric models have certain accuracy in estimation, their practical application faces multiple limitations. First, these methods heavily rely on high-performance hardware devices, such as high-resolution infrared cameras, depth sensors, and specialized eye trackers, significantly increasing system costs. Second, environmental conditions (e.g., low-light settings, strong reflections, or dynamic lighting) can severely disrupt the hardware’s ability to stably capture key ocular features (such as pupil edges and corneal reflex points), thereby reducing estimation reliability. Furthermore, due to the stringent hardware requirements and persistently high costs, such technologies struggle to gain traction in consumer-grade applications or large-scale real-world deployments.

### 2.2. Deep Learning Stage

At this stage, much work is based on deep learning, particularly convolutional neural networks (CNNs). Eye images and/or facial images are input into carefully designed networks. In 2015, Zhang et al. [14] proposed the first CNN-based algorithm for regressing gaze direction from left-eye images. In 2018, Fischer et al. [10] implemented a real-time algorithm involving CNNs with binocular images as input. Deng et al. [26] designed a transformation layer linking individual head poses and eye movements.

In addition to using eye images, some studies also use full-face images as network input. Zhang et al. [11] provided a spatial weighting mechanism that effectively encodes face position into a standard CNN architecture. GazeHF [27] combines inter-frame motion information with a lightweight model to match saccade and smooth pursuit phases. It can achieve high-precision and efficient eye tracking. Zhang et al. [28] extracted facial images and used mainstream lightweight networks and generative adversarial networks for gaze estimation.

Several studies have adopted multi-stream data inputs for their networks to enhance estimation accuracy. Professor Krafka from the University of Georgia collected the first large-scale eye-tracking dataset, GazeCapture [9], in 2016. It contains data from over 1450 people and was successfully used to train an eye-tracking network called iTracer, which takes full-face and binocular images as input. In 2020, Cheng et al. [29] proposed a coarse-to-fine network that first uses facial images to estimate basic gaze direction and then combines eye image residuals for fine correction. Mahmud et al. [19] proposed MSGazeNet, a gaze estimation method based on a multi-stream framework. This framework comprises two key components: a network designed to isolate anatomical eye regions and a second network for multi-stream gaze estimation. The generated eye region mask, combined with the original eye image, serves as a multi-stream input to the gaze estimation network.

Transformer was proposed by Vaswani et al. [30], initially achieving state-of-the-art performance in natural language processing tasks. Vision Transformer (ViT) [31] divides images into a series of patches and converts each patch into a vector representation as an input sequence. Specific visual tasks are accomplished by classifying or regressing the output of the Transformer encoder.

In recent years, Transformers have been introduced into gaze estimation tasks. GazeTR [32] considers two forms of Transformer for gaze inference: pure transformer and hybrid transformer. SCD-Net [33] introduced convolution and deconvolution self-attention in 2022, improving the generalization ability of gaze estimation and reducing costs. A multi-task gaze focus network [34] was proposed, along with four loss functions, to constrain the network in 2D and 3D spaces. Wu et al. [35] utilized the Transformer in Transformer (TNT) architecture to develop a gaze estimation model. The model captured both coarse-grained and fine-grained visual information pertaining to the face and, subsequently, integrated this information to provide feature representations for the estimation results. Xia et al. [36] implemented a parallel CNN Transformer aggregation network (CTA-Net) for gaze estimation, leveraging the global context of Transformer and the fact that CNN models excel in terms of details. Jindal et al. [37] proposed a framework that combines a spatial attention mechanism and a time series model. By using time series models, spatial observations can be transformed into temporal insights, thereby improving gaze estimation accuracy.

However, existing multi-stream models typically perform simple feature concatenation at the final stage when processing face and eye data, lacking deep interaction and fusion between features. This shallow integration fails to fully exploit the inherent correlations between facial and eye data, leading to suboptimal feature extraction. To address this limitation, we propose a novel framework that enables deep interaction and fusion between facial and eye features. There are two aspects: the interaction between two eye features and the interaction between eye and face features. Using this approach, our method significantly improves the accuracy and robustness of gaze estimation.

## 3. Method

Through integrating strategies such as multi-head attention mechanism, fusion, and interaction, the MAFI-Gaze (Multi-head Attention, Fusion, and Interaction Gaze) network for gaze estimation is proposed, in which facial and left and right eye images are input as multiple streams of data. There are multiple fusion modules between two of the eye images.

### 3.1. Network Structure

As shown in Figure 1, MAFI-Gaze consists of two main branches: one for extracting facial features and processing them interactively with eye features, and the other for extracting features from both eyes and fusing them. The first branch is connected to the second branch through the designed face and eye interaction (FEIM) module. The second branch consists mainly of the eye feature fusion module (EFFM), which focuses on the hierarchical fusion of the features of the two eyes through three fusion blocks (FBs). The output of the second branch, EFFM, is spliced with the output of the first branch, FEIM, to output the gaze estimation results using the multi-layer perceptron (MLP).

#### 3.1.1. Facial Feature Extraction Branch

First, facial feature extraction is performed through the top branch, as shown in Figure 1, sequentially executing the Layer1 and Layer2 modules. Layer1 and Layer2 are composed of stacked basic blocks, using different sizes of convolutional kernels to obtain different receptive fields. The basic blocks—namely, Block1, Block2, and Block3—are shown in Figure 2.

Block1 is used at the beginning of feature extraction. First, 3 × 3 depthwise separable convolution is used to extract features. Depthwise separable convolution independently convolves each input channel and then fuses the information between channels using 1 × 1 convolution. Then, the output of the convolutional layer is normalized using a Batch Normalization (BN) layer, which helps accelerate the training process and improve model stability. The SiLU activation function is used to help the network learn complex mapping relationships. Next, channel attention is used to weight the channels in the feature map, enhancing the feature representation of important channels and suppressing unimportant ones. Finally, 1 × 1 convolution is used to adjust the number of channels.

Based on Block1, Block2 adds a 1 × 1 convolution to adjust the number of channels, a BN layer to normalize the output of the convolutional layer, and the SiLU activation function. Based on Block2, Block3 adds DropPath and a shortcut connection. DropPath randomly drops some paths during training to mitigate the overfitting of the model.

Then, CBAM [38] is used to further learn the dependencies of different channels and spaces and enhance relevant features, as shown in Figure 3. CBAM combines channel attention and spatial attention to help the network better focus on important features, thereby improving model performance. The channel attention module generates channel attention weights through global pooling operations and a shared MLP. The spatial attention module generates spatial attention weights through channel-wise pooling and convolutional operations.

Through our experiments, we found that replacing maximum pooling with minimum pooling in the channel attention module and adding a shortcut connection allows the module to better capture detailed information in the feature map, thereby improving model performance. Unlike maximum pooling, which prioritizes dominant activations, our choice of global minimum pooling is driven by the need to preserve small yet critical ocular structures. For instance, pupils often exhibit weak feature responses, which are essential for accurate gaze direction inference. The shortcut connection retains the original feature information, avoiding potential information loss due to attention weights. The module can dynamically adjust feature weights according to task requirements while retaining the integrity of the original features.

As shown in Figure 4, the channel attention first undergoes average pooling and minimum pooling operations. These two pooling operations calculate the average and minimum values of the input data on each channel, respectively, to generate two feature maps with C × 1 × 1 dimensions. The convolutional layer is used to extract local features from the input data. Nonlinear activation functions introduce nonlinearity, and BN helps accelerate the training process and improve model stability. The sigmoid activation function limits the attention weights to between 0 and 1, facilitating subsequent weight allocation. Finally, a shortcut connection is used to prevent the loss of important information.

#### 3.1.2. Eye Feature Extraction and Fusion Branch

The eye feature extraction process and the eye feature fusion module (EFFM) are shown in Figure 5. First, low-level features are extracted from the input left/right eye images through the Layer1 convolutional layer. Then, the fusion block (FB) shown in Figure 6 is responsible for fusing the features of the left and right eye images, thereby utilizing the complementary information provided by both eyes. This is crucial for improving the accuracy and robustness of image processing.

The input to the FB comprises two features, X and Y, which are fused using multi-head cross-attention mechanisms and channel attention. The multi-head cross-attention mechanism combines cross-attention and multi-head attention mechanisms, as shown in Figure 7. It calculates the attention weights X-Y and Y-X between the two different inputs, capturing the interaction information between the two features, using multiple heads to capture features from different subspaces. Then, weights are assigned through channel attention, and the result is fused with the shortcut connection of X and Y.

The EFFM includes a total of three FBs. First, low-level features of both eyes are extracted by Layer1. Then, an FB is used to fuse them. Next, the fusion result is processed by Layer2. Finally, the output is fused with the results of processing both eyes through Layer3.

#### 3.1.3. Face and Eye Interaction Module

FEIM is designed to allow the facial features extracted from the top branch in Figure 1 to interact with the eye features fused by EFFM. As shown in Figure 8, FEIM fully utilizes the input facial features and left and right eye features to reallocate the weights of facial features, thereby guiding the facial features to the positions of both eyes.

EFFM performs GAP on the input features, concatenates the results, and sends them to the MLP. The result is fused with the input facial features to obtain the final features:(1)$$F_{face\text{-}out}=w*F_{face}$$
where ∗ denotes element-wise multiplication; $F_{face}$ represents the facial feature after the CA block of the top branch in Figure 1, and *w* is its corresponding re-assigned weight derived from(2)$$w=MLP\left(G\left(F_{face}\right)\;ⓒ\;G\left(F_{left}\right)\;ⓒ\;G\left(F_{right}\right)\right)$$
where MLP, *G*, and ⓒ stand for multi-layer perceptron (MLP), global average pooling (GAP), and feature concatenation, respectively; and $F_{left}$ and $F_{right}$ represent the left and right eye features output by EFFM.

#### 3.1.4. Gaze Estimation Result

At the output end of the network, the three features obtained from the interaction module (FEIM) and the eye feature fusion module (EFFM) are concatenated. Finally, the final gaze estimation is output through MLP: (3)$$Gaze\text{-}out=MLP\left(F_{face\text{-}out}\;ⓒ\;F_{left\text{-}out}\;ⓒ\;F_{right\text{-}out}\right)$$
where MLP represents the multi-layer perceptron and $F_{left\text{-}out}$ and $F_{right\text{-}out}$ represent the output features of the left and right eyes of EFFM in Figure 1, respectively.

### 3.2. Loss Function

Here, both Mean Absolute Error (MAE) and Inverse Cosine Loss (arccos) are used to supervise gaze prediction, expressed, respectively, as follows:(4)$$L_{MAE}=\frac{1}{n}\sum \left|g-\hat{g}\right|$$(5)$$L_{arccos}=arccos\cdot \frac{g\cdot \hat{g}}{∥g∥\cdot ∥\hat{g}∥}$$
where $\hat{g}$ and *g* represent the predicted gaze position expressed by Equation (Equation 3) and the actual gaze, respectively.

Our overall loss function is designed as(6)$$L_{total}=\lambda_{1}L_{MAE}+\lambda_{2}L_{arccos}$$
where $\lambda_{1}$ and $\lambda_{2}$ are hyper-parameters.

## 4. Experiments

### 4.1. Datasets

We summarize the three publicly available gaze estimation datasets—Gaze360, MPIIFaceGaze, and EYEDIAP—in Table 1. The distribution of gaze directions for these datasets is shown in Figure 9.

Gaze360 [13]: The Gaze360 dataset includes 172K images from 238 participants, covering a variety of head poses and gaze distributions, various indoor and outdoor capture environments, and various subjects. In this dataset, 129K images, 17K images, and 26K images are used for training, validation, and evaluation, respectively.

MPIIFaceGaze [15]: The MPIIFaceGaze dataset is based on the MPIIGaze dataset. It was collected from the real world and contains 45K photos from 15 individuals. The dataset covers a variety of lighting conditions. In addition, we trained MPIIFaceGaze using the leave-one-person-out strategy, taking an average of 15 sessions.

EYEDIAP [20]: The EYEDIAP dataset primarily constitutes data from RGB and RGB-D cameras, controlling variables such as head pose, individual participants, environment, and perceptual conditions. The dataset consists of 94 videos from 16 subjects, with a total duration of 237 min. Referring to the preprocessing method of [39], we randomly divided all subjects into four clusters and implemented the leave-one-out assessment strategy in these four clusters.

We employed the data rectification method proposed in [15,39] to preprocess the data, effectively mitigating environmental disturbances. All facial and eye images underwent image equalization. Their width and height are standardized to 224 pixels, and the size of the eye image is 36 × 60.

### 4.2. Evaluation Indicator

Performance is measured using the mean gaze angular error. The gaze angle is first converted into the 3D Cartesian coordinate system:(7)$$\nu_{g}=T\left(g\right)=\left[{-\cos\emptyset }_{p}{\sin\emptyset }_{y},{-\sin\emptyset }_{p},\;{-\cos\emptyset }_{p}{\sin\emptyset }_{y}\right]$$(8)$$\nu_{\hat{g}}=T\left(\hat{g}\right)=\left[{-\cos\emptyset }_{\hat{p}}{\sin\emptyset }_{\hat{y}},{-\sin\emptyset }_{\hat{p}},\;{-\cos\emptyset }_{\hat{p}}{\sin\emptyset }_{\hat{y}}\right]$$
where $T\left(\cdot \right)$ represents the transformation between two coordinate systems. Then, the gaze angular error denoted by δ between the real gaze, *g*, and the predicted gaze, $\hat{g}$, is calculated:(9)$$\delta =arccos\cdot \frac{\nu_{g}^{T}\cdot \nu_{\hat{g}}}{∥\nu_{g}∥\cdot ∥\hat{\nu_{g}}∥}$$

### 4.3. Implementation Details

MAFI-Gaze is implemented using the Pytorch platform. It is optimized by the Adam optimizer with a learning rate of 0.0005 and a batch size of 32; the total number of epochs is 100. We used a linear learning rate warmup, which was set at five epochs. The learning rate decreased by a factor of 0.5, and the decay step was set as 20 epochs. The dropout probability for the fully connected layer was set at 0.2.

### 4.4. Experimental Results

Table 2 shows the usage of different methods for three types of inputs: facial image, eye images, and head pose. Facial image is used as input for most methods. Some methods (e.g., RT-Gene, GazeNet, Dilated-Net, etc.) specifically use eye images as input. In addition to accepting facial or eye images, only a few methods (e.g., GazeNet and Dilated-Net) consider head pose as additional input information. Our proposed MAFI-Gaze uses facial and eye images as input to the network.

Table 2 shows a comparison between MAFI-Gaze and other state-of-the-art methods using the same training and testing sets. On the MPIIFaceGaze and EYEDIAP datasets, MAFI-Gaze achieves mean angular errors of 3.84° and 5.01°, respectively, ranking first. On the Gaze360 dataset, MAFI-Gaze achieves a mean angular error of 10.78°, ranking in the middle. Hybrid-SAM ranks first on Gaze360 but lower than our method on the EYEDIAP dataset. Considering the above, our method achieves satisfactory gaze estimation results.

As shown in Figure 10, we conducted a comparative analysis of gaze estimation errors across individual subjects in the MPIIFaceGaze dataset. The experimental results reveal two key findings: (1) MAFI-Gaze achieves superior accuracy, outperforming Dilated-Net in 11 out of 15 subjects (73.3%) and surpassing CTA-Net in 8 cases (53.3%), demonstrating robust performance despite significant inter-subject appearance variations; (2) stability is improved, with MAFI-Gaze exhibiting the smallest standard deviation (0.75) compared to Dilated-Net (0.91) and CTA-Net (0.79), suggesting better consistency in handling diverse facial characteristics.

We compared the average angular errors of MAFI-Gaze and GazeTR-Hybirdacross for different gaze angle ranges on the Gaze360 dataset, as illustrated in Figure 11. The results reveal the following observations: (1) In the yaw direction, MAFI-Gaze achieves lower errors than GazeTR-Hybird for large angles (e.g., 90–180 degrees); however, such samples constitute a small proportion of the dataset. In contrast, MAFI-Gaze underperforms GazeTR-Hybird in the remaining angular ranges (−90 to 90 degrees). (2) In the pitch direction, MAFI-Gaze outperforms GazeTR-Hybird in the range of −10 to 10 degrees, and such samples account for more than 50% of the total samples. In other angular ranges, GazeTR-Hybird performs better than MAFI-Gaze. In summary, MAFI-Gaze excels in large yaw angles and small pitch angles but falls short in other ranges. On the other hand, GazeTR-Hybird delivers more consistent performance across most angular ranges, particularly in medium yaw angles and large pitch angles.

In addition, as shown in Figure 12, we calculated the average gray value of images on the Gaze360 dataset and compared the corresponding average angular errors. The average gray value represents the average intensity of all pixels in an image, where a higher value (closer to 255) indicates a brighter image. In comparison, a lower value (closer to 0) indicates a darker image. The average gray value is a simple and intuitive metric suitable for quickly assessing the overall brightness of an image. By analyzing the impact of image brightness on prediction performance, we observe the following: (1) Within the average gray value range of 20–40, which contains the largest proportion of samples, MAFI-Gaze achieves lower errors than GazeTR-Hybird; (2) within the range of 40–100, GazeTR-Hybird outperforms MAFI-Gaze; (3) in darker or brighter environments (extremely low or high gray values), MAFI-Gaze demonstrates better performance than GazeTR-Hybird. However, in the intermediate brightness range, GazeTR-Hybird exhibits superior performance.

Table 3 shows the performance comparison of channel attention methods using different pooling methods on the three datasets (Gaze360, MPIIFaceGaze, and EYEDIAP). There are four methods: minimum pooling, average pooling, maximum pooling, and shortcut connections. Using only average pooling for channel attention yields moderate performance on Gaze360 and MPIIFaceGaze but performs the worst on EYEDIAP. Adding maximum pooling to average pooling improves performance on EYEDIAP but reduces performance on Gaze360 and MPIIFaceGaze. Adding minimum pooling to average pooling improves the results on all three datasets. Furthermore, adding shortcut connections to minimum pooling and average pooling improves performance on all three datasets. In conclusion, combining minimum pooling, average pooling, and shortcut connections yields the best results.

### 4.5. Ablation Experiments

A series of ablation experiments were conducted to verify that each proposed module or branch is indispensable and beneficial. In terms of branches, (1) “MAFI-Gaze w/o face” and (2) “MAFI-Gaze w/o eyes” use only two eye images (excluding facial images) or facial images (excluding eye images) as input, respectively. Additionally, for all blocks including, FEIM and EFFM, (3) “MAFI-Gaze w/o FEIM” and (4) “MAFI-Gaze w/o EFFM” indicate the removal of FEIM and the fusion blocks in EFFM from the network, respectively, whereas (5) the proposed “MAFI-Gaze” fully utilizes all these modules.

The results are shown in Table 4. It is evident that using only facial or eye images as input increases the gaze angular error. Conversely, if both are input as multi-stream data, the angular error is significantly reduced, demonstrating the usefulness of both branches for facial and eye images. Furthermore, it is clear that the integration of EFFM and fusion blocks in EFFM greatly reduces the error. Overall, these blocks significantly improve gaze estimation accuracy while only slightly increasing computational load and network size.

We evaluated the impact of two hyper-parameters, $\lambda_{1}$ and $\lambda_{2}$, from the loss function on the network’s performance. Since $\lambda_{1}$ is the weight for the Mean Absolute Error (MAE) and $\lambda_{1}$ is the weight for the angular error, there is a significant difference in their magnitudes. We initially set $\lambda_{1}$ to 50 and $\lambda_{2}$ to 1 to address this. The results are detailed in Table 5, which shows that increasing $\lambda_{1}$ to 100 improves performance across all three datasets. However, further increasing $\lambda_{1}$ to 150 leads to a decline in performance.

## 5. Discussion

We developed an eye feature fusion module leveraging multi-head cross-attention; this facilitates interaction among eye features during the feature extraction phase. Additionally, we designed a facial feature interaction module that emphasizes features related to the eyes. Furthermore, we improved the channel attention mechanism by replacing maximum pooling with minimum pooling and incorporating a shortcut; this enable the model to focus more effectively on fine-grained details. This enhancement collectively contributes to improving the model’s performance.

Our model exhibits superior performance on the MPIIFaceGaze and EYEDIAP datasets but only moderate performance on the Gaze360 dataset. Consequently, we compared our method with GazeTR-Hybird based on gaze angle (yaw/pitch) and image brightness (represented by the average gray value). The results revealed that while our method performs well at large angles, it falls short at intermediate angles. Similarly, our method does not perform as strongly with images of intermediate brightness. Therefore, our process should improve the robustness of the model rather than focusing too much on details, perhaps leading to increased noise.

While our multi-branch design and attention-based fusion modules significantly improve accuracy, they also introduce additional computational costs and parameter counts compared to simpler or single-stream approaches. This trade-off may limit the real-time deployment of our method on resource-constrained devices, indicating the need for future optimization of models for resource-constrained environments such as edge devices.

## 6. Conclusions

This study proposed MAFI-Gaze for the prediction of 3D gaze direction in unconstrained environments. Facial images and left and right eye images are used as multi-stream input for MAFI-Gaze. We designed a fusion block that incorporates multi-head cross-attention and channel attention. The eye branches achieve direct interaction and effective feature fusion during extraction through the designed fusion block. Additionally, we designed a face-and-eye interaction module to combine the extracted facial features with the eye features obtained from the interaction module. This integration helps to guide facial features towards the eye regions, leading to superior estimation results. Through experiments, we found that replacing maximum pooling with minimum pooling in the CBAM channel attention module and incorporating a shortcut connection enables the module to better capture detailed information in the feature map, subsequently improving the model’s performance. Comparative experiments and ablation studies on the Gaze360, MPIIFaceGaze, and EYEDIAP datasets demonstrate the effectiveness of MAFI-Gaze.

## Figures and Tables

**Figure 1 sensors-25-01893-f001:**
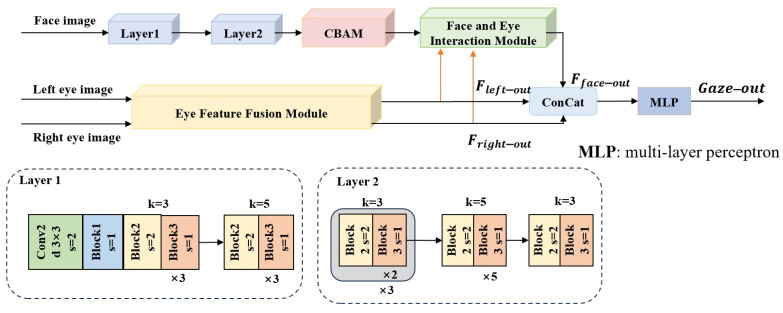
Architecture of the proposed network: MAFI-Gaze.

**Figure 2 sensors-25-01893-f002:**
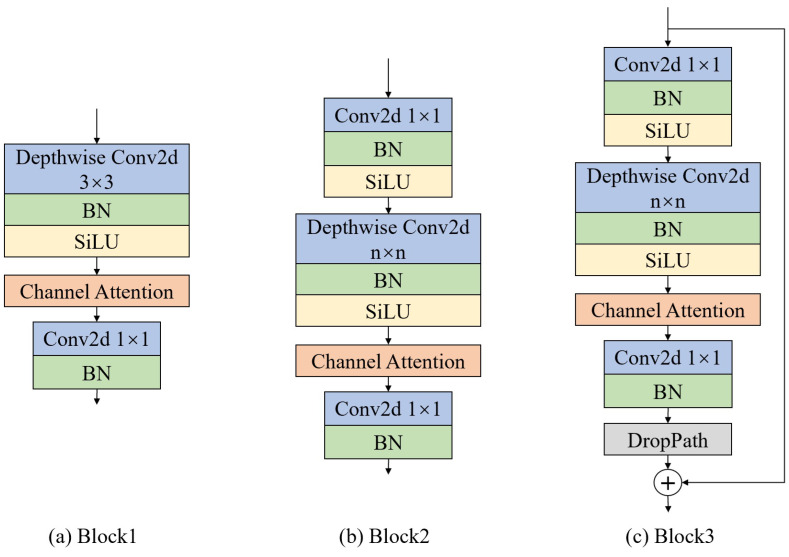
Basic blocks.

**Figure 3 sensors-25-01893-f003:**
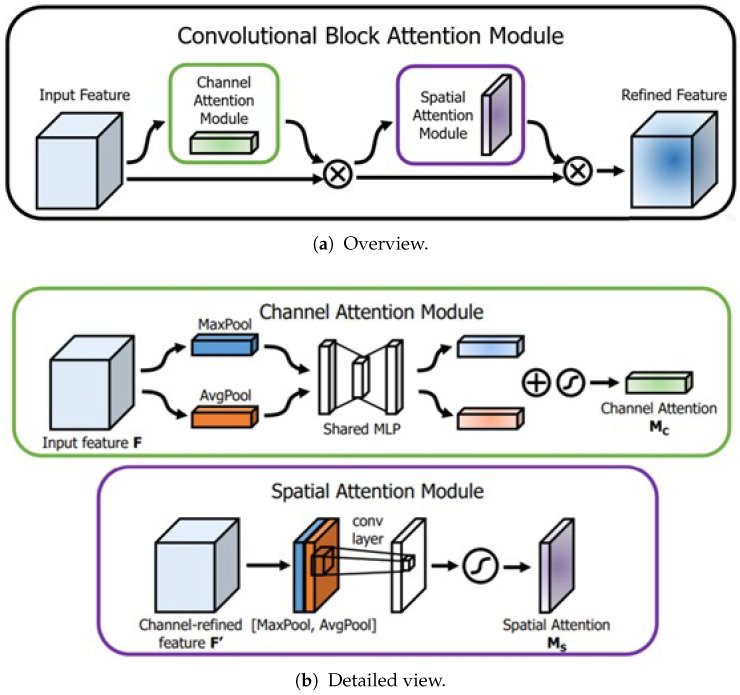
CBAM [38].

**Figure 4 sensors-25-01893-f004:**
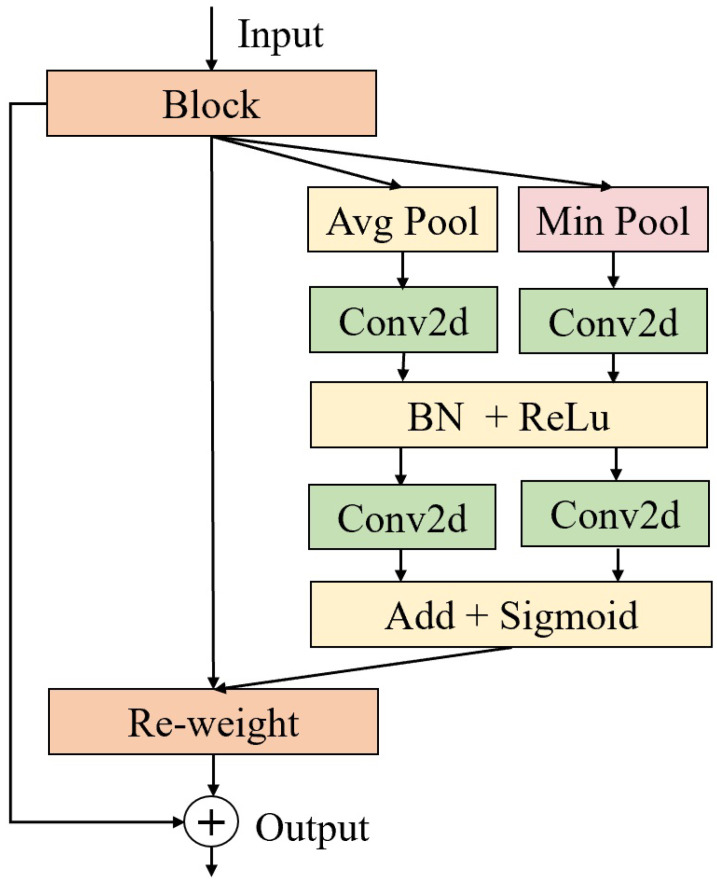
Improved channel attention.

**Figure 5 sensors-25-01893-f005:**
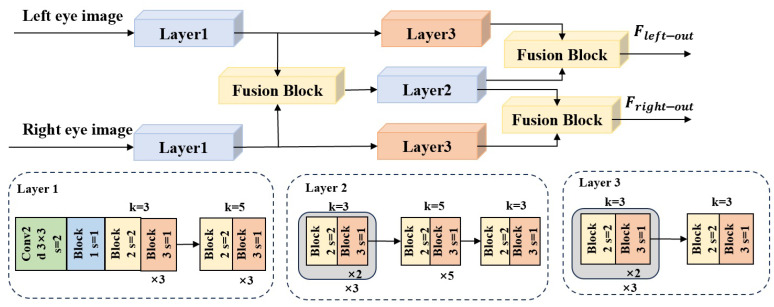
Eye feature fusion module (EFFM).

**Figure 6 sensors-25-01893-f006:**
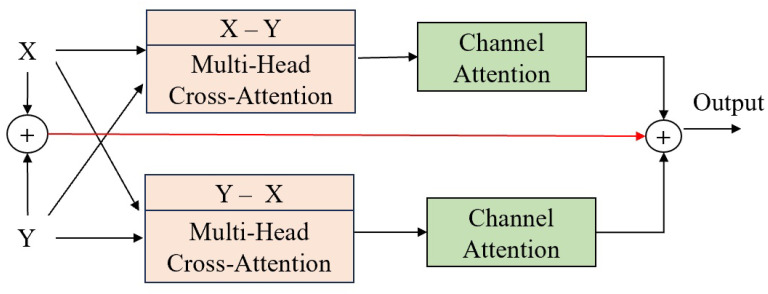
Fusion block (FB).

**Figure 7 sensors-25-01893-f007:**
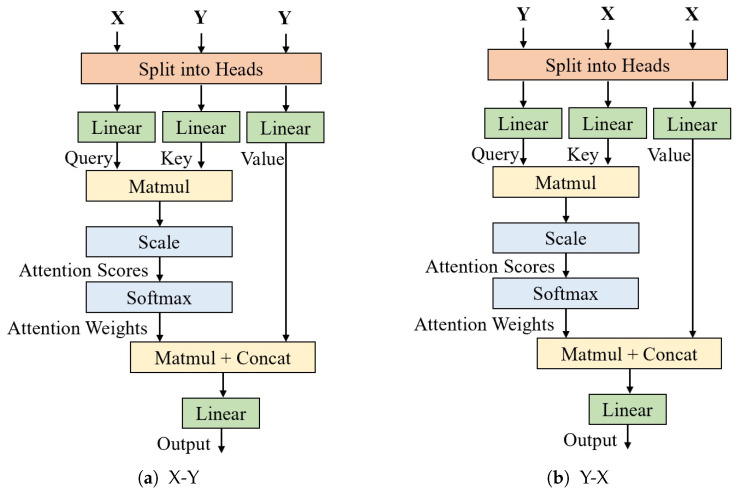
Face and eye interaction module (FEIM).

**Figure 8 sensors-25-01893-f008:**
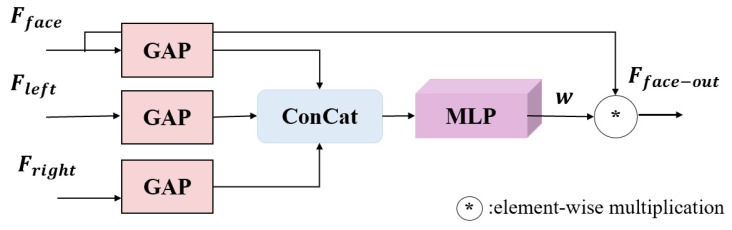
Face and eye interaction module (FEIM).

**Figure 9 sensors-25-01893-f009:**
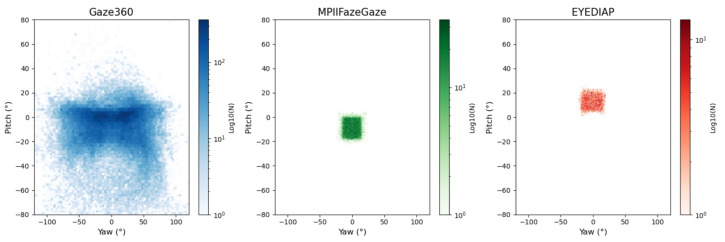
Distribution of gaze angle in different datasets.

**Figure 10 sensors-25-01893-f010:**
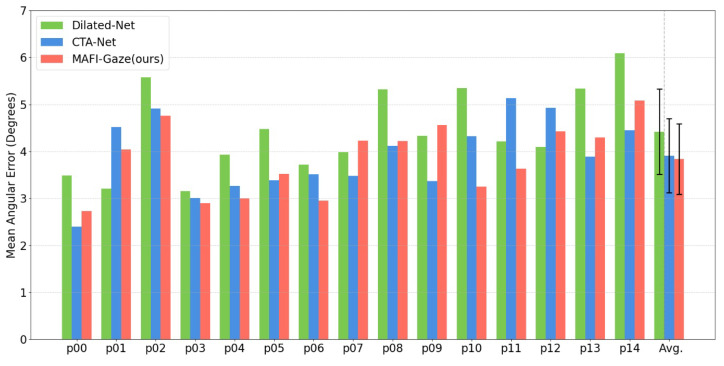
Comparison of mean angular error among different subjects on the MPIIFaceGaze dataset. Error bars indicate standard deviations computed across subjects.

**Figure 11 sensors-25-01893-f011:**
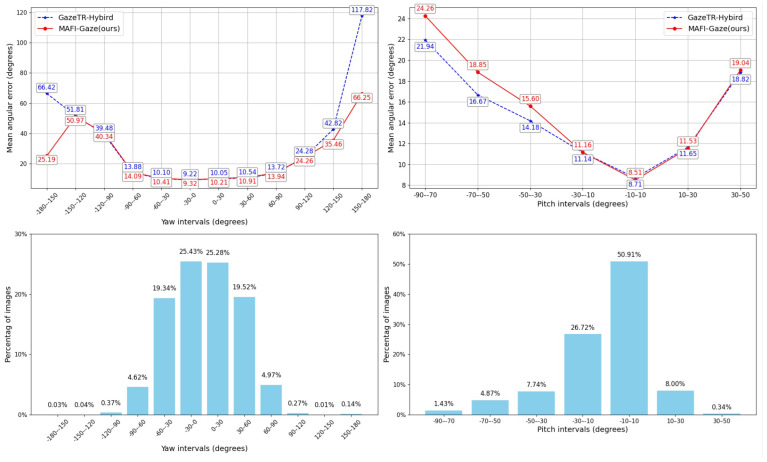
Comparison of mean angular error in different gaze angle ranges on the Gaze360 dataset.

**Figure 12 sensors-25-01893-f012:**
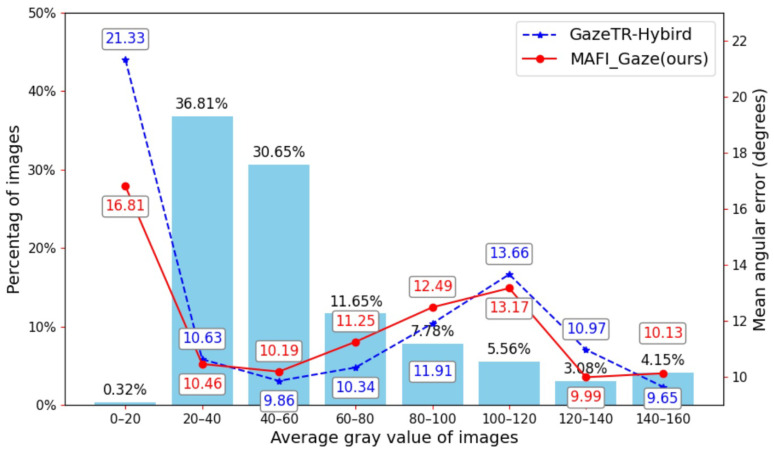
Comparison of mean angular error in different average grayscale values on the Gaze360 dataset.

**Table 1 sensors-25-01893-t001:** Dataset comparison.

Feature Dimension	Gaze360	MPIIFaceGaze	EYEDIAP
Data collection environment	Complex outdoor	Lab-controlled	Dynamic interaction
Head pose range	±70°	±30°	±45°
Illumination variation	Strong light/shadow	Artificial light	Indoor natural light
Sample diversity	238 participants	15 participants	16 scenarios

**Table 2 sensors-25-01893-t002:** Comparison between MAFI-Gaze and other methods.

Method	Input			Angular Error		
	Face image	Eye Images	Head Pose	Gaze360	MPIIFaceGaze	EYEDIAP
FullFace [11]	*√*			14.99°	4.93°	6.53°
RT-Gene [10]		*√*		12.26°	4.66°	6.02°
GazeNet [15]		*√*	*√*	-	5.76°	6.79°
Dilated-Net [16]	*√*	*√*	*√*	13.37	4.42°	6.19°
Gaze360 [13]	*√*			11.04°	4.06°	5.36°
CA-Net [29]	*√*	*√*		11.20°	4.27°	5.27°
GazeTR-Pure [32]	*√*			13.58°	4.74°	5.72°
GazeTR-Hybird [32]	*√*			10.62°	4.00°	5.17°
SCD-Net [33]	*√*			10.70°	4.04°	5.25°
GFNet [34]	*√*	*√*		-	3.96°	5.40°
GTiT-Hybrid [35]	*√*			11.20°	4.11°	5.35°
CTA-Net [36]	*√*	*√*		10.44°	3.91°	-
MSGazeNet [19]		*√*		-	4.64°	5.86°
Hybrid-SAM [37]	*√*			10.05°	-	6.54°
MAFI-Gaze (ours)	*√*	*√*		10.78°	3.84°	5.01°

**Table 3 sensors-25-01893-t003:** Comparison of channel attention methods for MAFI-Gaze.

Pooling			Shortcut	Gaze360	MPIIFaceGaze	EYEDIAP
Minimum	Average	Maximum				
	*√*	*√*		11.47°	4.26°	5.04°
	*√*	*√*	*√*	11.41°	4.22°	5.21°
*√*		*√*		11.31°	4.01°	5.13°
	*√*			11.09°	3.94°	5.28°
*√*	*√*			11.02°	3.92°	5.14°
*√*	*√*		*√*	10.78°	3.84°	5.01°

**Table 4 sensors-25-01893-t004:** MAFI-Gaze ablation experiment results.

Method	Gaze360	MPIIFaceGaze	EYEDIAP	FLOPs (G)	Params (M)
MAFI-Gaze w/o face	13.12°	4.60°	5.53°	0.21	6.21
MAFI-Gaze w/o eyes	11.07°	4.14°	5.25°	1.34	6.67
MAFI-Gaze w/o FEI	10.91°	4.08°	5.13°	1.56	12.89
MAFI-Gaze w/o EFF	10.85°	3.92°	5.06°	1.50	12.88
MAFI-Gaze	10.78°	3.84°	5.01°	1.56	12.90

**Table 5 sensors-25-01893-t005:** Evaluation of hyper-parameters in the loss function.

$\lambda_{1}$	$\lambda_{2}$	Gaze360	MPIIFaceGaze	EYEDIAP
50	1	11.07°	4.17°	5.24°
100	1	10.78°	3.84°	5.01°
150	1	10.90°	3.88°	4.97°

## Data Availability

The data that support the findings of this study are openly available at https://gaze360.csail.mit.edu/ (accessed on 28 December 2023), https://www.idiap.ch/en/scientific-research/data/eyediap (accessed on 27 December 2023), and https://www.perceptualui.org/research/datasets/MPIIFaceGaze/ (accessed on 19 January 2024).

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
