# Peer review of "Gaze Estimation Network Based on Multi-Head Attention, Fusion, and Interaction"

_sensors, 2025, doi:10.3390/s25061893_

Round 1
Reviewer 1 Report
Comments and Suggestions for Authors
See attached file.

Reviewer 2 Report
Comments and Suggestions for Authors
The manuscript presents a MAFI-Gaze network leveraging multi-head attention and feature interaction for gaze estimation, demonstrating innovation in multi-stream architectures. While the experimental results on MPIIFaceGaze and EYEDIAP datasets show state-of-the-art performance. However, there are several significant issues that need to be addressed. Therefore, MAJOR revision is required before this manuscript could be accepted for publication.
Comment 1. The authors need to provide detailed information on current progress in gaze estimation, including a broader review of existing methods and their limitations. This will help position the proposed framework within the context of the current research landscape.
Comment 2. The comparison methods chosen are not currently the most optimal in terms of performance. It is recommended that the author include more mainstream methods for comparison in order to validate the performance level of the proposed method.
Comment 3.The tuning strategies for λ₁and λ₂are not explained in the main text. It is recommended to add ablation experiments in the experimental section to verify the impact of these two hyperparameters on network performance.
Comment 4.The datasets chosen by the author are relatively outdated. It is recommended to select the latest publicly available datasets to further validate the experimental results.
Comment 5.The author has implemented a gaze estimation network using fusion and interaction strategies, but the advantages of these strategies have not been adequately demonstrated. It is recommended that the author provide visualizations to highlight the strengths of the proposed network. For example, visualizations could include comparisons of gaze estimation accuracy, error reduction, or other performance metrics between the proposed network and other methods. Additionally, the author could consider using the latest publicly available datasets to further validate the experimental results.
Comments on the Quality of English Language
The current manuscript needs to be polished by a native English speaker or a professional language editing service to ensure clarity and readability.
Round 2
Reviewer 2 Report
Comments and Suggestions for Authors
The authors have revised and improved the manuscript according to the comments raised last time.